# Learning to Optimize in Swarms

**Yue Cao, Tianlong Chen, Zhangyang Wang, Yang Shen**
Departments of Electrical and Computer Engineering & Computer Science and Engineering
Texas A&M University, College Station, TX 77840
{cyppsp,wiwjp619,atlaswang,yshen}@tamu.edu

## Abstract

Learning to optimize has emerged as a powerful framework for various optimization
and machine learning tasks. Current such "meta-optimizers" often learn in the
space of continuous optimization algorithms that are point-based and uncertainty-
unaware. To overcome the limitations, we propose a meta-optimizer that learns
in the algorithmic space of both point-based and population-based optimization
algorithms. The meta-optimizer targets at a meta-loss function consisting of both
cumulative regret and entropy. Specifically, we learn and interpret the update
formula through a population of LSTMs embedded with sample- and feature-level
attentions. Meanwhile, we estimate the posterior directly over the global optimum
and use an uncertainty measure to help guide the learning process. Empirical results
over non-convex test functions and the protein-docking application demonstrate
that this new meta-optimizer outperforms existing competitors. The codes are
publicly available at: `https://github.com/Shen-Lab/LOIS`.

## 1   Introduction

Optimization provides a mathematical foundation for solving quantitative problems in many fields,
along with numerical challenges. The *no free lunch* theorem indicates the non-existence of a univer-
sally best optimization algorithm for all objectives. To manually design an effective optimization
algorithm for a given problem, many efforts have been spent on tuning and validating pipelines,
architectures, and hyperparameters. For instance, in deep learning, there is a gallery of gradient-based
algorithms specific to high-dimensional, non-convex objective functions, such as Stochastic Gradient
Descent [1], RmsDrop [2], and Adam [3]. Another example is in *ab initio* protein docking whose
energy functions as objectives have extremely rugged landscapes and are expensive to evaluate.
Gradient-free algorithms are thus popular there, including Markov chain Monte Carlo (MCMC) [4]
and Particle Swarm Optimization (PSO) [5].

To overcome the laborious manual design, an emerging approach of meta-learning (learning to learn)
takes advantage of the knowledge learned from related tasks. In meta-learning, the goal is to learn
a meta-*learner* that could solve a set of problems, where each sample in the training or test set is a
particular problem. As in classical machine learning, the fundamental assumption of meta-learning
is the generalizability from solving the training problems to solving the test ones. For optimization
problems, a key to meta-learning is how to efficiently utilize the information in the objective function
and explore the space of optimization algorithms.

In this study, we introduce a novel framework in meta-learning, where we train a meta-optimizer that
learns in the space of both point-based and population-based optimization algorithms for continuous
optimization. To that end, we design a novel architecture where a population of RNNs (specifically,
LSTMs) jointly learn iterative update formula for a population of samples (or a swarm of particles).
To balance exploration and exploitation in search, we directly estimate the posterior over the optimum
and include in the meta-loss function the differential entropy of the posterior. Furthermore, we
embed feature- and sample-level attentions in our meta-optimizer to interpret the learned optimization

strategies. Our numerical experiments, including global optimization for nonconvex test functions and an application of protein docking, endorse the superiority of the proposed meta-optimizer.

## 2 Related work

Meta-learning originated from the field of psychology [6, 7]. [8, 9, 10] optimized a learning rule in a parameterized learning rule space. [11] used RNN to automatically design a neural network architecture. More recently, learning to learn has also been applied to sparse coding [12, 13, 14, 15], plug-and-play optimization [16], and so on.

In the field of learning to optimize, [17] proposed the first framework where gradients and function values were used as the features for RNN. A coordinate-wise structure of RNN relieved the burden from the enormous number of parameters, so that the same update formula was used independently for each coordinate. [18] used the history of gradients and objective values as states and step vectors as actions in reinforcement learning. [19] also used RNN to train a meta-*learner* to optimize black-box functions, including Gaussian process bandits, simple control objectives, and hyper-parameter tuning tasks. Lately, [20] introduced a hierarchical RNN architecture, augmented with additional architectural features that mirror the known structure of optimization tasks.

The target applications of previous methods are mainly focused on training deep neural networks, except [19] focusing on optimizing black-box functions. There are three **limitations** of these methods. First, they learn in a limited algorithmic space, namely point-based optimization algorithms that use gradients or not (including SGD and Adam). So far there is no method in learning to learn that reflects population-based algorithms (such as evolutionary and swarm algorithms) proven powerful in many optimization tasks. Second, their learning is guided by a limited meta loss, often the cumulative regret in sampling history that primarily drives exploitation. One exception is the expected improvement (EI) used by [19] under Gaussian processes. Last but not the least, these methods do not interpret the process of learning update formula, despite the previous usage of attention mechanisms in [20].

To overcome aforementioned limitations of current learning-to-optimize methods, we present a new meta-optimizer with the following **contributions**:

- (*Where to learn*): We learn in an extended space of both point-based and population-based optimization algorithms;
- (*How to learn*): We incorporate the posterior into meta-loss to guide the search in the algorithmic space and balance the exploitation-exploration trade-off.
- (*What more to learn*): We design a novel architecture where a population of LSTMs jointly learn iterative update formula for a population of samples and embedded sample- and feature-level attentions to explain the formula.

## 3 Method

### 3.1 Notations and background

We use the following convention for notations throughout the paper. Scalars, vectors (column vectors unless stated otherwise), and matrices are denoted in lowercase, bold lowercase, and bold uppercase, respectively. Superscript $'$ indicates vector transpose.

Our goal is to solve the following optimization problem:

$$\boldsymbol{x}^* = \arg \min_{\boldsymbol{x} \in \mathbb{R}^n} f(\boldsymbol{x}).$$

Iterative optimization algorithms, either point-based or population-based, have the same generic update formula:

$$\boldsymbol{x}^{t+1} = \boldsymbol{x}^t + \delta \boldsymbol{x}^t,$$

where $\boldsymbol{x}^t$ and $\delta \boldsymbol{x}^t$ are the sample (or a single sample called "particle" in swarm algorithms) and the update (a.k.a. step vector) at iteration $t$, respectively. The update is often a function $g(\cdot)$ of historic sample values, objective values, and gradients. For instance, in point-based gradient descent,

$$\delta \boldsymbol{x}^t = g(\{\boldsymbol{x}^\tau, f(\boldsymbol{x}^\tau), \nabla f(\boldsymbol{x}^\tau)\}_{\tau=1}^t) = -\alpha \nabla f(\boldsymbol{x}^t),$$

where $\alpha$ is the learning rate. In particle swarm optimization (PSO), assuming that there are $k$ samples (particles), then for particle $j$, the update is determined by the entire population:

$$\delta \boldsymbol{x}_j^t = g(\{\{\boldsymbol{x}_j^\tau, f(\boldsymbol{x}_j^\tau), \nabla f(\boldsymbol{x}_j^\tau)\}_{j=1}^k\}_{\tau=1}^t) = w\delta\boldsymbol{x}_j^{t-1} + r_1(\boldsymbol{x}_j^t - \boldsymbol{x}_j^{t*}) + r_2(\boldsymbol{x}_j^t - \boldsymbol{x}^{t*}),$$

where $\boldsymbol{x}_j^{t*}$ and $\boldsymbol{x}^{t*}$ are the best position (with the smallest objective value) of particle $j$ and among all particles, respectively, during the first $t$ iterations; and $w, r_1, r_2$ are the hyper-parameters often randomly sampled from a fixed distribution (e.g. standard Gaussian distribution) during each iteration.

In most of the modern optimization algorithms, the update formula $g(\cdot)$ is analytically determined and fixed during the whole process. Unfortunately, similar to what the *No Free Lunch Theorem* suggests in machine learning, there is no single best algorithm for all kinds of optimization tasks. Every state-of-art algorithm has its own best-performing problem set or domain. Therefore, it makes sense to learn the optimal update formula $g(\cdot)$ from the data in the specific problem domain, which is called "learning to optimize". For instance, in [17], the function $g(\cdot)$ is parameterized by a recurrent neural network (RNN) with input $\nabla f(\boldsymbol{x}^t)$ and the hidden state from the last iteration: $g(\cdot) = \text{RNN}(\nabla f(\boldsymbol{x}^t), \boldsymbol{h}^{t-1})$. In [19], the inputs of RNN are $\boldsymbol{x}^t, f(\boldsymbol{x}^t)$ and the hidden state from the last iteration: $g(\cdot) = \text{RNN}(\boldsymbol{x}^t, f(\boldsymbol{x}^t), \boldsymbol{h}^{t-1})$.

### 3.2 Population-based learning to optimize with posterior estimation

We describe the details of our population-based meta-optimizer in this section. Compared to previous meta-optimizers, we employ $k$ samples whose update formulae are learned from the population history and are individually customized, using attention mechanisms. Specifically, our update rule for particle $i$ could be written as:

$$g_i(\cdot) = \text{RNN}_i\left(\alpha_i^{\text{inter}}(\{\alpha_j^{\text{intra}}(\{\boldsymbol{S}_j^\tau\}_{\tau=1}^t)\}_{j=1}^k), \boldsymbol{h}_i^{t-1}\right)$$

where $\boldsymbol{S}_j^t = (\boldsymbol{s}_{j1}^t, \boldsymbol{s}_{j2}^t, \boldsymbol{s}_{j3}^t, \boldsymbol{s}_{j4}^t)$ is a $n \times 4$ feature matrix for particle $j$ at iteration $t$, $\alpha_j^{\text{intra}}(\cdot)$ is the intra-particle attention function for particle $j$, and $\alpha_i^{\text{inter}}(\cdot)$ is the $i$-th output of the inter-particle attention function. $\boldsymbol{h}_i^{t-1}$ is the hidden state of the $i$th LSTM at iteration $t-1$.

For typical population-based algorithms, the same update formula is adopted by all particles. We follow the convention to set $g_1(\cdot) = g_2(\cdot) = ... = g_k(\cdot)$, which suggests $\text{RNN}_i = \text{RNN}$ and $\alpha_j^{\text{intra}}(\cdot) = \alpha^{\text{intra}}(\cdot)$.

We will first introduce the feature matrix $\boldsymbol{S}_j^\tau$ and then describe the intra- and inter- attention modules.

### 3.2.1 Features from different types of algorithms

Considering the expressiveness and the searchability of the algorithmic space, we consider the update formulae of both point- and population-based algorithms and choose the following four features for particle $i$ at iteration $t$:

- gradient: $\nabla f(\boldsymbol{x}_i^t)$
- momentum: $\boldsymbol{m}_i^t = \sum_{\tau=1}^t (1-\beta)\beta^{t-1}\nabla f(\boldsymbol{x}_i^\tau)$
- velocity: $\boldsymbol{v}_i^t = \boldsymbol{x}_i^t - \boldsymbol{x}_i^{t*}$
- attraction: $\frac{\sum_j \left(\exp(-\alpha d_{ij}^2)(\boldsymbol{x}_i^t - \boldsymbol{x}_j^t)\right)}{\sum_j \exp(-\alpha d_{ij}^2)}$, for all $j$ that $f(\boldsymbol{x}_j^t) < f(\boldsymbol{x}_i^t)$. $\alpha$ is a hyperparameter and $d_{ij} = ||\boldsymbol{x}_i^t - \boldsymbol{x}_j^t||_2$.

These four features include two from point-based algorithms using gradients and the other two from population-based algorithms. Specifically, the first two are used in gradient descent and Adam. The third feature, velocity, comes from PSO, where $\boldsymbol{x}_i^{t*}$ is the best position (with the lowest objective value) of particle $i$ in the first $t$ iterations. The last feature, attraction, is from the Firefly algorithm [21]. The attraction toward particle $i$ is the weighted average of $\boldsymbol{x}_i^t - \boldsymbol{x}_j^t$ over all $j$ such that $f(\boldsymbol{x}_j^t) < f(\boldsymbol{x}_i^t)$; and the weight of particle $j$ is the Gaussian similarity between particle $i$ and $j$. For the particle of the smallest $f(\boldsymbol{x}_i^t)$, we simply set this feature vector to be zero. In this paper, we use $\beta = 0.9$ and $\alpha = 1$.

It is noteworthy that each feature vector is of dimension $n \times 1$, where $n$ is the dimension of the search space. Besides, the update formula in each base-algorithm is linear w.r.t. its corresponding feature. To learn a better update formula, we will incorporate those features into our model of deep neural networks, which is described next.

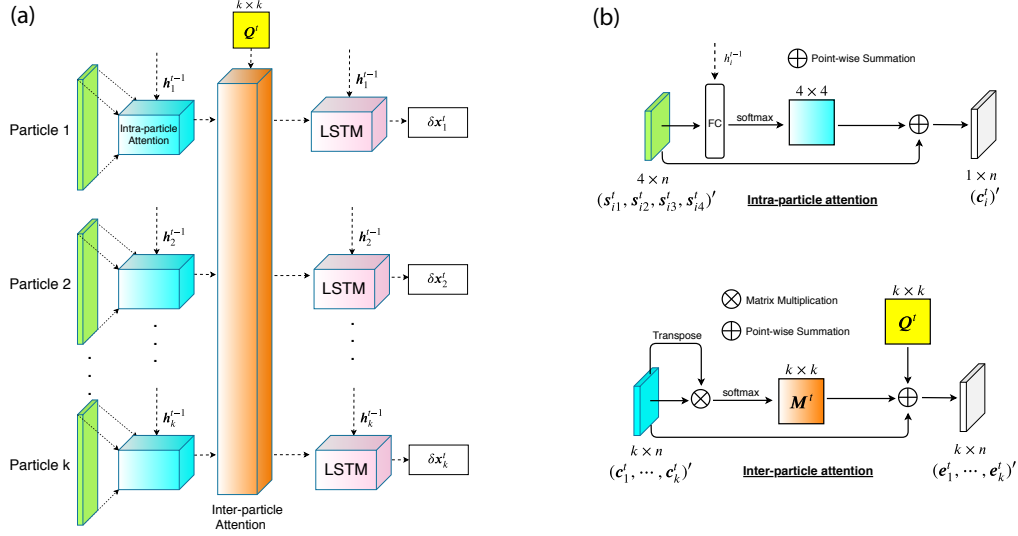

Figure 1: (a) The architecture of our meta-*optimizer* for one step. We have $k$ particles here. For each particle, we have gradient, momentum, velocity and attraction as features. Features for each particle will be sent into an intra-particle (feature-level) attention module, together with the hidden state of the previous LSTM. The outputs of $k$ intra-particle attention modules, together with a kernelized pairwise similarity matrix $\boldsymbol{Q}^t$ (yellow box in the figure), will be the input of an inter-particle (sample-level) attention module. The role of inter-particle attention module is to capture the cooperativeness of all particles in order to reweight features and send them into $k$ LSTMs. The LSTM's outputs, $\delta\boldsymbol{x}$, will be used for generating new samples. (b) The architectures of intra- and inter-particle attention modules.

### 3.2.2 Overall model architecture

Fig. 1a depicts the overall architecture of our proposed model. We use a population of LSTMs and design two attention modules here: feature-level ("intra-particle") and sample-level ("inter-particle") attentions. For particle $i$ at iteration $t$, the intra-particle attention module is to reweight each feature based on the context vector $\boldsymbol{h}_i^{t-1}$, which is the hidden state from the $i$-th LSTM in the last iteration. The reweight features of all particles are fed into an inter-particle attention module, together with a $k \times k$ distance similarity matrix. The inter-attention module is to learn the information from the rest $k-1$ particles and affect the update of particle $i$. The outputs of inter-particle attention module will be sent into $k$ identical LSTMs for individual updates.

### 3.2.3 Attention mechanisms

For the intra-particle attention module, we use the idea from [22, 23, 24]. As shown in Fig. 1b, given that the $j$th input feature of the $i$th particle at iteration $t$ is $\boldsymbol{s}_{ij}^t$, we have:

$$b_{ij}^t = \boldsymbol{v}_a^T \tanh\left(\boldsymbol{W}_a \boldsymbol{s}_{ij}^t + \boldsymbol{U}_a \boldsymbol{h}_{ij}^t\right), \quad p_{ij}^t = \frac{\exp(b_{ij}^t)}{\sum_{r=1}^4 \exp(b_{ir}^t)},$$

where $\boldsymbol{v}_a \in \mathbb{R}^n$, $\boldsymbol{W}_a \in \mathbb{R}^{n \times n}$ and $\boldsymbol{U}_a \in \mathbb{R}^{n \times n}$ are the weight matrices, $\boldsymbol{h}_i^{t-1} \in \mathbb{R}^n$ is the hidden state from the $i$th LSTM in iteration $t-1$, $b_{ij}^t$ is the output of the fully-connected (FC) layer and $p_{ij}^t$ is the output after the softmax layer. We then use $p_{ij}^t$ to reweight our input features:

$$\boldsymbol{c}_i^t = \sum_{r=1}^4 p_{ir}^t \boldsymbol{s}_{ir}^t,$$

where $\boldsymbol{c}_i^t \in \mathbb{R}^n$ is the output of the intra-particle attention module for the $i$th particle at iteration $t$.

For inter-particle attention, we model $\delta\boldsymbol{x}_i^t$ for each particle $i$ under the impacts of the rest $k-1$ particles. Specific considerations are as follows.

- The closer two particles are, the more they impact each other's update. Therefore, we construct a kernelized pairwise similarity matrix $\boldsymbol{Q}^t \in \mathbb{R}^{k \times k}$ (column-normalized) as the weight matrix. Its element is $q_{ij}^t = \frac{exp(-\frac{||\boldsymbol{x}_i^t - \boldsymbol{x}_j^t||^2}{2l})}{\sum_{r=1}^{k} exp(-\frac{||\boldsymbol{x}_r^t - \boldsymbol{x}_j^t||^2}{2l})}$.

- The similar two particles are in their intra-particle attention outputs ($\boldsymbol{c}_i^t$, local suggestions for updates), the more they impact each other's update. Therefore, we introduce another weight matrix $\boldsymbol{M}^t \in \mathbb{R}^{k \times k}$ whose element is $m_{ij} = \frac{\exp\left((\boldsymbol{c}_i^t)'\boldsymbol{c}_j^t\right)}{\sum_{r=1}^{k} \exp\left((\boldsymbol{c}_r^t)'\boldsymbol{c}_j^t\right)}$ (normalized after column-wise softmax).

As shown in Fig. 1b, the output of the inter-particle module for the $j$th particle will be:

$$\boldsymbol{e}_j^t = \gamma \sum_{r=1}^{k} m_{rj}^t q_{rj}^t \boldsymbol{c}_r^t + \boldsymbol{c}_j^t,$$

where $\gamma$ is a hyperparameter which controls the ratio of contribution of rest $k$-1 particles to the $j$th particle. In this paper, $\gamma$ is set to be 1 without further optimization.

### 3.2.4 Loss function, posterior estimation, and model training

Cumulative regret is a common meta loss function: $L(\boldsymbol{\phi}) = \sum_{t=1}^{T} \sum_{j=1}^{k} f(\boldsymbol{x}_j^t)$. However, this loss function has two main drawbacks. First, the loss function does not reflect any exploration. If the search algorithm used for training the optimizer does not employ exploration, it can be easily trapped in the vicinity of a local minimum. Second, for population-based methods, this loss function tends to drag all the particles to quickly converge to the same point.

To balance the exploration-exploitation tradeoff, we bring the work from [25] — it built a Bayesian posterior distribution over the global optimum $\boldsymbol{x}^*$ as $p(\boldsymbol{x}^*| \bigcup_{t=1}^{T} D_t)$, where $D_t$ denotes the samples at iteration $t$: $D_t = \left\{ \left(\boldsymbol{x}_j^t, f(\boldsymbol{x}_j^t)\right) \right\}_{j=1}^{k}$. We claim that, in order to reduce the uncertainty about the whereabouts of the global minimum, the best next sample can be chosen to minimize the entropy of the posterior, $h\left(p(\boldsymbol{x}^*| \bigcup_{t=1}^{T} D_t)\right)$. Therefore, we propose a loss function for function $f(\cdot)$ as:

$$\ell_f(\boldsymbol{\phi}) = \sum_{t=1}^{T} \sum_{j=1}^{k} f(\boldsymbol{x}_j^t) + \lambda h\left(p(\boldsymbol{x}^*| \bigcup_{t=1}^{T} D_t)\right),$$

where $\lambda$ controls the balance between exploration and exploitation and $\boldsymbol{\phi}$ is a vector of model parameters.

Following [25], the posterior over the global optimum is modeled as a Boltzmann distribution:

$$p(\boldsymbol{x}^*| \bigcup_{t=1}^{T} D_t) \propto \exp(-\rho \hat{f}(\boldsymbol{x})),$$

where $\hat{f}(\boldsymbol{x})$ is a function estimator and $\rho$ is the annealing constant. In the original work of [25], both $\hat{f}(\boldsymbol{x})$ and $\rho$ are updated over iteration $t$ for active sampling. In our work, they are fixed since the complete training sample paths are available at once.

Specifically, for a function estimator based on samples in $\bigcup_{t=1}^{T} D_t$, we use a Kriging regressor [26] which is known to be the best unbiased linear estimator (BLUE):

$$\hat{f}(\boldsymbol{x}) = f_0(\boldsymbol{x}) + (\boldsymbol{\kappa}(\boldsymbol{x}))' (\boldsymbol{K} + \epsilon^2 \boldsymbol{I})^{-1}(\boldsymbol{y} - \boldsymbol{f_0}),$$

where $f_0(\boldsymbol{x})$ is the prior for $\mathrm{E}[f(\boldsymbol{x})]$ (we use $f_0(\boldsymbol{x}) = 0$ in this study); $\boldsymbol{\kappa}(\boldsymbol{x})$ is the kernel vector with the $i$th element being the kernel, a measure of similarity, between $\boldsymbol{x}$ and $\boldsymbol{x}_i$; $\boldsymbol{K}$ is the kernel matrix with the $(i,j)$-th element being the kernel between $\boldsymbol{x}_i$ and $\boldsymbol{x}_j$; $\boldsymbol{y}$ and $\boldsymbol{f_0}$ are the vector consisting of $y_1, \ldots, y_{n_t}$ and $f_0(\boldsymbol{x}_1), \ldots, f_0(\boldsymbol{x}_{n_t})$, respectively; and $\epsilon$ reflects the noise in the observation and is often estimated to be the average training error (set at 2.1 in this study).

For $\rho$, we follow the annealing schedule in [25] with one-step update:

$$\rho = \rho_0 \cdot \exp\left((h_0)^{-1}\left|\bigcup_{t=1}^{T} D_t\right|^{\frac{1}{n}}\right),$$

where $\rho_0$ is the initial parameter of $\rho$ ($\rho_0 = 1$ without further optimization here); $h_0$ is the initial entropy of the posterior with $\rho = \rho_0$; and $n$ is the dimensionality of the search space.

In total, our meta loss for $m$ functions $f_q(\cdot)$ ($q = 1, \dots, m$) (analogous to $m$ training examples) with L2 regularization is

$$L(\boldsymbol{\phi}) = \frac{1}{m}\sum_{q=1}^{m}\ell_{f_q}(\boldsymbol{\phi}) + C||\boldsymbol{\phi}||_2^2.$$

To train our model we use the optimizer Adam which requires gradients. The first-order gradients are calculated numerically through TensorFlow following [17]. We use coordinate-wise LSTM to reduce the number of parameters. In our implementation the length of LSTM is set to be 20. For all experiments, the optimizer is trained for 10,000 epochs with 100 iterations in each epoch.

## 4 Experiments

We test our meta-optimizer through convex quadratic functions, non-convex test functions and an optimization-based application with extremely noisy and rugged landscapes: protein docking.

### 4.1 Learn to optimize convex quadratic functions

In this case, we are trying to minimize a convex quadratic function:

$$f(\boldsymbol{x}) = ||\boldsymbol{A}_q\boldsymbol{x} - \boldsymbol{b}_q||_2^2,$$

where $\boldsymbol{A}_q \in \mathbb{R}^{n \times n}$ and $\boldsymbol{b}_q \in \mathbb{R}^{n \times 1}$ are parameters, whose elements are sampled from i.i.d. normal distributions for the training set. We compare our algorithm with SGD, Adam, PSO and DeepMind's LSTM (DM_LSTM) [17]. Since different algorithms have different population sizes, for fair comparison we fix the total number of objective function evaluations (sample updates) to be 1,000 for all methods. The population size $k$ of our meta-optimizer and PSO is set to be 4, 10 and 10 in the 2D, 10D and 20D cases, respectively. During the testing stage, we sample another 128 pairs of $\boldsymbol{A}_q$ and $\boldsymbol{b}_q$ and evaluate the current best function value at each step averaged over 128 functions. We repeat the procedure 100 times in order to obtain statistically significant results.

As seen in Fig. 2, our meta-optimizer performs better than DM_LSTM in the 2D, 10D, and 20D cases. Both meta-optimizers perform significantly better than the three baseline algorithms (except that PSO had similar convergence in 2D).

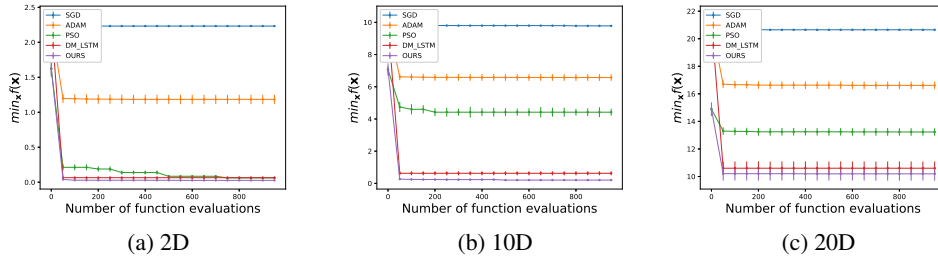

| (a) 2D | (b) 10D | (c) 20D |

Figure 2: The performance of different algorithms for quadratic functions in (a) 2D, (b) 10D, and (c) 20D. The mean and the standard deviation over 100 runs are evaluated every 50 function evaluations.

We also compare our meta-optimizer's performances with and without the guiding posterior in meta loss. As shown in the supplemental Fig. S1, including the posterior improves optimization performances especially in higher dimensions. Meanwhile, posterior estimation in higher dimensions presents more challenges. The impact of posteriors will be further assessed in ablation study.

## 4.2 Learn to optimize non-convex Rastrigin functions

We then test the performance on a non-convex test function called Rastrigin function:

$$f(\boldsymbol{x}) = \sum_{i=1}^{n} x_i^2 - \sum_{i=1}^{n} \alpha \cos\left(2\pi x_i\right) + \alpha n,$$

where $\alpha = 10$. We consider a broad family of similar functions $f_q(\boldsymbol{x})$ as the training set:

$$f_q(\boldsymbol{x}) = ||\boldsymbol{A}_q \boldsymbol{x} - \boldsymbol{b}_q||_2^2 - \alpha \boldsymbol{c}_q \cos(2\pi \boldsymbol{x}) + \alpha n, \qquad (1)$$

where $\boldsymbol{A}_q \in \mathbb{R}^{n \times n}$, $\boldsymbol{b}_q \in \mathbb{R}^{n \times 1}$ and $\boldsymbol{c}_q \in \mathbb{R}^{n \times 1}$ are parameters whose elements are sampled from i.i.d. normal distributions. It is obvious that Rastrigin is a special case in this family with $\boldsymbol{A} = \boldsymbol{I}, \boldsymbol{b} = \{0, 0, \ldots, 0\}', \boldsymbol{c} = \{1, 1, \ldots, 1\}'$.

During the testing stage, 100 i.i.d. trajectories are generated in order to reach statistically significant conclusions. The population size $k$ of our meta-optimizer and PSO is set to be 4, 10 and 10 for 2D, 10D and 20D, respectively. The results are shown in Fig. 3. In the 2D case, our meta-optimizer and PSO perform fairly the same while DM_LSTM performs much worse. In the 10D and 20D cases, our meta-optimizer outperforms all other algorithms. It is interesting that PSO is the second best among all algorithms, which indicates that population-based algorithms have unique advantages here.

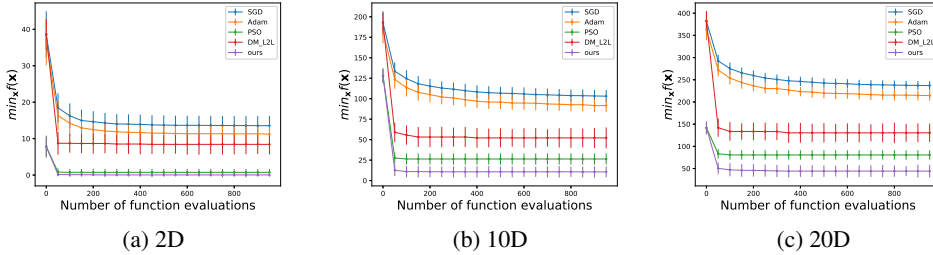

| (a) 2D | (b) 10D | (c) 20D |

Figure 3: The performance of different algorithms for a Rastrigin function in (a) 2D, (b) 10D, and (c) 20D. The mean and the standard deviation over 100 runs are evaluated every 50 function evaluations.

## 4.3 Transferability: Learning to optimize non-convex functions from convex optimization

We also examine the transferability from convex to non-convex optimization. The hyperparameter $\alpha$ in Rastrigin family controls the level of ruggedness for training functions: $\alpha = 0$ corresponds to a convex quadratic function and $\alpha = 10$ does the rugged Rastrigin function. Therefore, we choose three different values of $\alpha$ (0, 5 and 10) to build training sets and test the resulting three trained models on the 10D Rastrigin function. From the results in the supplemental Fig. S2, our meta-optimizer's performances improve when it is trained with increasing $\alpha$. The meta-optimizer trained with $\alpha = 0$ had limited progress over iterations, which indicates the difficulty to learn from convex functions to optimize non-convex rugged functions. The one trained with $\alpha = 5$ has seen significant improvement.

## 4.4 Interpretation of learned update formula

In an effort to rationalize the learned update formula, we choose the 2D Rastrigin test function to illustrate the interpretation analysis. We plot sample paths of our algorithm, PSO and Gradient Descent (GD) in Fig 4a. Our algorithm finally reaches the funnel (or valley) containing the global optimum ($\boldsymbol{x} = \boldsymbol{0}$), while PSO finally reaches a suboptimal funnel. At the beginning, samples of our meta-optimizer are more diverse due to the entropy control in the loss function. In contrast, GD is stuck in a local minimum which is close to its starting point after 80 samples.

To further show which factor contributes the most to each update, we plot the feature weight distribution over the first 20 iterations. Since for particle $i$ at iteration $t$, the output of its intra-attention module is a weighted sum of its 4 features: $\boldsymbol{c}_i^t = \sum_{r=1}^{4} p_{ir}^t \boldsymbol{s}_{ir}^t$, we hereby sum $p_{ir}^t$ for the $r$-th feature over all particles $i$. The final weight distribution (normalized) over 4 features reflecting the contribution of each feature at iteration $t$ is shown in Fig. 4b. In the first 6 iterations,

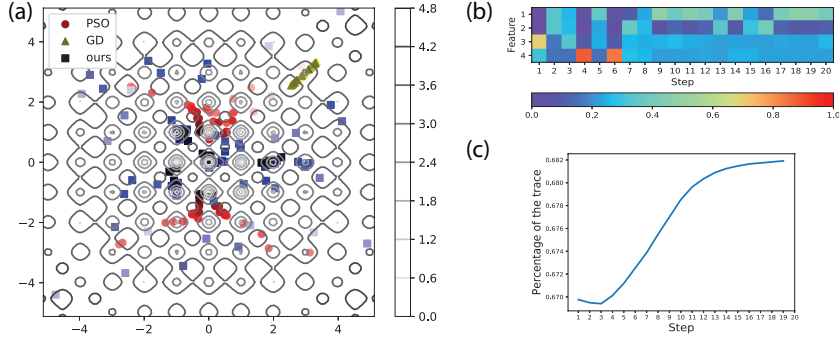

Figure 4: (a) Paths of the first 80 samples of our meta-optimizer, PSO and GD for 2D Rastrigin functions. Darker shades indicate newer samples. (b) The feature attention distribution over the first 20 iterations for our meta-optimizer. (c) The percentage of the trace of $\gamma \boldsymbol{Q}^t \odot \boldsymbol{M}^t + \boldsymbol{I}$ (reflecting self-impact on updates) over iteration $t$.

the population-based features contribute to the update most. Point-based features start to play an important role later.

Finally, we examine in the inter-particle attention module the level of particles working collaboratively or independently. In order to show this, we plot the percentage of the diagonal part of $\gamma \boldsymbol{Q}^t \odot \boldsymbol{M}^t + \boldsymbol{I}$: $\frac{\text{tr}(\gamma \boldsymbol{Q}^t \odot \boldsymbol{M}^t + \boldsymbol{I})}{\sum \gamma \boldsymbol{Q}^t \odot \boldsymbol{M}^t + \boldsymbol{I}}$ ($\odot$ denotes element-wise product), as shown in Fig. 4c. It can be seen that, at the beginning, particles are working more collaboratively. With more iterations, particles become more independent. However, we note that the trace (reflecting self impacts) contributes 67%-69% over iterations and the off-diagonals (impacts from other particles) do above 30%, which demonstrates the importance of collaboration, a unique advantage of population-based algorithms.

## 4.5 Ablation study

How and why our algorithm outperforms DM_LSTM is both interesting and important to unveil the underlying mechanism of the algorithm. In order to deeply understand each part of our algorithms, we performed an ablation study to progressively show each part's contribution. Starting from the DM_LSTM baseline ($\boldsymbol{B}_0$), we incrementally crafted four algorithms: running DM_LSTM for $k$ times under different initializations and choosing the best solution ($\boldsymbol{B}_1$); using $k$ independent particles, each with the two point-based features, the intra-particle attention module, and the hidden state ($\boldsymbol{B}_2$); adding the two population-based features and the inter-particle attention module to $\boldsymbol{B}_2$ so as to convert $k$ independent particles into a swarm ($\boldsymbol{B}_3$); and eventually, adding an entropy term in meta loss to $\boldsymbol{B}_3$, resulting in our **Proposed** model.

We tested the five algorithms ($\boldsymbol{B}_0$–$\boldsymbol{B}_3$ and the **Proposed**) on 10D and 20D Rastrigin functions with the same settings as in Section 4.2. We compare the function minimum values returned by these algorithms in the table below (reported are means $\pm$ standard deviations over 100 runs, each using 1,000 function evaluations).

| Dimension | $\boldsymbol{B}_0$ | $\boldsymbol{B}_1$ | $\boldsymbol{B}_2$ | $\boldsymbol{B}_3$ | **Proposed** |
|---|---|---|---|---|---|
| 10 | 55.4±13.5 | 48.4±10.5 | 40.1±9.4 | 20.4±6.6 | 12.3±5.4 |
| 20 | 140.4±10.2 | 137.4±12.7 | 108.4±13.4 | 48.5±7.1 | 43.0 ±9.2 |

Our key observations are as follows. i) $\boldsymbol{B}_1$ v.s. $\boldsymbol{B}_0$: their performance gap is marginal, which proves that our performance gain is not simply due to having $k$ independent runs; ii) $\boldsymbol{B}_2$ v.s. $\boldsymbol{B}_1$ and $\boldsymbol{B}_3$ v.s. $\boldsymbol{B}_2$: Whereas including intra-particle attention in $\boldsymbol{B}_2$ already notably improves the performance compared to $\boldsymbol{B}_1$, including population-based features and inter-particle attention in $\boldsymbol{B}_3$ results in the largest performance boost. This confirms that our algorithm majorly benefits from the attention mechanisms; iii) **Proposed** v.s. $\boldsymbol{B}_3$: adding entropy from the posterior gains further, thanks to its balancing exploration and exploitation during search.

### 4.6 Application to protein docking

We bring our meta-*optimizer* into a challenging real-world application. In computational biology, the structural knowledge about how proteins interact each other is critical but remains relatively scarce [27]. Protein docking helps close such a gap by computationally predicting the 3D structures of protein-protein complexes given individual proteins' 3D structures or 1D sequences [28]. *Ab initio* protein docking represents a major challenge of optimizing a noisy and costly function in a high-dimensional conformational space [25].

Mathematically, the problem of *ab initio* protein docking can be formulated as optimizing an extremely rugged energy function: $f(\boldsymbol{x}) = \Delta G(\boldsymbol{x})$, the Gibbs binding free energy for conformation $\boldsymbol{x}$. We calculate the energy function in a CHARMM 19 force field as in [5] and shift it so that $f(\boldsymbol{x}) = 0$ at the origin of the search space. And we parameterize the search space as $\mathbb{R}^{12}$ as in [25]. The resulting $f(\boldsymbol{x})$ is fully differentiable in the search space. For computational concern and batch training, we only consider 100 interface atoms. We choose a training set of 25 protein-protein complexes from the protein docking benchmark set 4.0 [29] (see Supp. Table S1 for the list), each of which has 5 starting points (top-5 models from ZDOCK [30]). In total, our training set includes 125 instances. During testing, we choose 3 complexes (with 1 starting model each) of different levels of docking difficulty. For comparison, we also use the training set from Eq. 1 ($n = 12$). All methods including PSO and both versions of our meta-optimizer have $k = 10$ particles and 40 iterations in the testing stage.

As seen in Fig. 5, both meta-optimizers achieve lower-energy predictions than PSO and the performance gains increase as the docking difficulty level increases. The meta-optimizer trained on other protein-docking cases performs similarly as that trained on the Rastrigin family in the easy case and outperforms the latter in the difficult case.

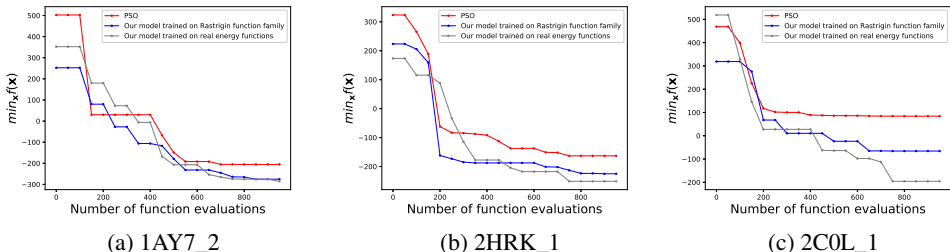

| (a) 1AY7_2 | (b) 2HRK_1 | (c) 2C0L_1 |

Figure 5: The performance of PSO, our meta-optimizer trained on Rastrigin function family and that trained on real energy functions for three different levels of docking cases: (a) rigid (easy), (b) medium, and (c) flexible (difficult).

## 5 Conclusion

Designing a well-behaved optimization algorithm for a specific problem is a laborious task. In this paper, we extend point-based meta-*optimizer* into population-based meta-*optimizer*, where update formulae for a sample population are jointly learned in the space of both point- and population-based algorithms. In order to balance exploitation and exploration, we introduce the entropy of the posterior over the global optimum into the meta loss, together with the cumulative regret, to guide the search of the meta-*optimizer*. We further embed intra- and inter- particle attention modules to interpret each update. We apply our meta-optimizer to quadratic functions, Rastrigin functions and a real-world challenge – protein docking. The empirical results demonstrate that our meta-optimizer outperforms competing algorithms. Ablation study shows that the performance improvement is directly attributable to our algorithmic innovations, namely population-based features, intra- and inter-particle attentions, and posterior-guided meta loss.

#### Acknowledgments

This work is in part supported by the National Institutes of Health (R35GM124952 to YS). Part of the computing time is provided by the Texas A&M High Performance Research.

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
