[Supplementary Material]

# Learning to Optimize in Swarms
## (Supporting Information)

**Yue Cao, Tianlong Chen, Zhangyang Wang, Yang Shen**

Departments of Electrical and Computer Engineering & Computer Science and Engineering
Texas A&M University, College Station, TX 77840
{cyppsp,wiwjp619,atlaswang,yshen}@tamu.edu

# 1 Supplemental Results

## 1.1 Learn to optimize convex quadratic functions

Figure S1: The performance of our meta-optimizer for convex quadratic functions, with or without the posterior term in meta loss.

## 1.2 Transferability: Learning to optimize non-convex Rastrigin functions from convex optimization

Figure S2: Optimization performance for the 10D Rastrigin function ($\alpha = 10$) when the meta-optimizer is trained using Rastrigin family with increasing $\alpha$ (thus increasing ruggedness). $\alpha = 0$ corresponds to convex training functions.

## 2 Training Examples for Protein Docking

| Difficulty level | Protein Data Bank (PDB) code |
|---|---|
| Rigid | 1N8O , 7CEI , 1DFJ , 1AVX , 1BVN , 1IQD , 1CGI , 1MAH , 1EZU , 1JPS , 1PPE , 1R0R , 2I25 , 2B42 , 1EAW , 2JEL , 1BJ1 , 1KXQ , 1EWY |
| Medium | 1XQS, 1M10, 1IJK,1GRN |
| Flexible | 1IBR, 1ATN |

Table S1: 4-letter ID of proteins used in the training set.