[Reviews · NeurIPS 2019]

Reviewer 1



### Overall opinion * While this method is useful for certain non-convex problems, I’m not sure if neurips is a suitable publication venue for this paper. ### Clarity * The motivation for each feature is well laid out. * The overall motivation is clear and the design of the model reflects this. ### Originality * To the best of my knowledge, this is the first work to propose a meta-optimizer for population-based optimization. * The entropy term that encourages exploration seems novel, at least in the context of optimizer learning. ### Significance * The model performs well on a variety of benchmarks. * While this paper seems to be a useful contribution to highly non convex optimization problems such as protein docking, it is a fairly straightforward combination of optimizer learning and population-based optimization. ------------------------ post-rebuttal ---------------------------- The author response and other reviews have convinced me of the significance of this work; I change my score from 5 to 6.

Reviewer 2



This paper introduces a new meta-learning algorithm that combines population-based and point-based optimization. While population based approaches have been very popular in very rugged landscapes, current meta-learning methods are point-based and thus not suitable for optimizing such functions. This work presents two contributions, (1) a new architecture for population based meta-learning. This architecture, while more complicated, can be summarized as follows: each particle is composed of a set of 4 features (gradient, momentum, velocity, and attractions), an attention mechanism is applied to those features together with the hidden state. The outputs of the attention mechanism for all particles are fed into an inter-particle attention together with a similarity matrix. The output of this inter-attention mechanism constitutes the intput of the LSTM learner that outputs the variation on the optimized parameters. The second contribution is the addition of a differential entropy term on the meta-loss that balances exploration and exploitation of the optimization process. This paper tackles the important problem of extending current meta-learning algorithms to take advantage of population-based training, which is necessary in extremely non-convex problems. I consider that the contributions of their work are novel, especially the proposed architecture. While the work is well motivated, the paper lacks clarity. More specifically, it leaves a lot of important components to the supplementary material to the point that it is impossible to fully understand the approach without reading the supplementary material. The paper could be restructured so all of it fits in the 8 pages limit without compromising readability. In this line, my main concerns in the paper are: How is the P(x*| D_t) defined? It seems a crucial part of your contribution, but the paper lacks its definition (besides citing Chao and She, 2019) The model architecture should be more clearly explained in the main section. A key component of your approach, is the attention mechanism it seems crucial to me that you explain in the main text how this works. Right now, your main contribution it’s explained in just a paragraph. Section 4.1, heavily discusses plots and results from the supp material. Those results are interesting and important, they should be included in the main body. Figure 3 ©, the definition of Q and M should be in the main text, otherwise it’s impossible to interpret what’s that plot means without looking at the supplementary material. Section 4.4 results are entirely in to supp material. Those results are interesting and important, they should be included in the main body. Regarding the experimental evaluation, the paper would highly benefit of an ablation study. This work presents an architecture that consists of many parts, however it is not clear which parts have significant effects. Regarding baselines, an important baseline to run would be to run the DM_LSTM for k different initializations and pick the best. This would show if your method just benefits from having k independent runs or there’s actually a benefit in the attention mechanisms (given that the particle interdependence is not high).

Reviewer 3



The authors did an outstanding job in addressing "where to learn" "how to learn" "what more to learn" and clearly established the novelty of their method. Their work opens the door to solving more sophisticated optimization using L2L. The uncertainty-aware loss fits the goal of the exploration-exploitation tradeoff, that was popularly researched in Bayesian optimization and RL (but not so much yet in L2L). I also like the intra- and inter- particle attention modules, that add to the explainability whether particles are working collaboratively or independently. A clear conclusion could be drawn from their Rastrigin experiments, that population-based L2L outperforms gradient-based meta optimizers including (Andrychowicz et al., 2016). The meta optimizer then outperformed a very recent SOTA method (Cao and Shen, 2019) in the real protein docking application.

[Author Response · NeurIPS 2019]

We thank all three reviewers for unanimously recognizing the significance and merits of our work. We have addressed
all their raised concerns below. And we promise to release all codes and pre-trained models upon acceptance.

**Structure and Clarity (R2).** We thank **R2** for pointing out the important issue. Despite that R2 considers our
contributions as significant, we agree with R2 that "*it needs a clearer explanation...*", and further "*the paper could be*
*restructured so all of it fits in the 8 pages limit without compromising readability*".

Our concrete action plan to re-organize the existing materials is as follows:

• First, we will merge Fig. 1 (two attention mechanisms) from Supporting Information (**SI**) into Fig. 1 in the main
text. Accordingly, we will elaborate on the details of model architectures, including the matrices $Q$ and $M$, in
Section 3.2.2 of the main text rather than Section 1 of **SI**. In addition, we will annotate important notations in the
new Fig. 1. In this way we will make model architectures and our first contribution (population-based meta learning)
more organized and more clarified as suggested by R2.
• Second, we will describe the posterior distribution, $P(x^*|D_t)$, in Section 3.2.3 of the main text. This could make
our second contribution (differential entropy in meta loss) more clear to the readers.
• Third, we will move Fig. 2 from **SI** into the main text as Fig. 2(d)–(f) to better explain results in Section 4.1.
• Fourth, we will move Fig. 3 from **SI** into the main text to clearly explain transferability results in Section 4.4.

We will make space in the main text for the above, by moving the pseudo code of Algorithm 1 and some details about
protein docking experiments (Section 4.5) into **SI**. We have already prepared a preliminary version with those revisions.

**Ablation Study (R2).** We deeply acknowledge the valuable suggestion. To elucidate "which parts have significant
effects", we performed an ablation study to progressively show each part's contribution. Starting from the DM_LSTM
baseline ($\mathbf{B}_0$), we incrementally crafted four models: running DM_LSTM for $k$ times under different initializations and
choosing the best solution ($\mathbf{B}_1$); using $k$ independent particles, each with the two point-based features, the intra-particle
attention module, and the hidden state ($\mathbf{B}_2$); adding the two population-based features and the inter-particle attention
module to $\mathbf{B}_2$ so as to convert $k$ independent particles into a swarm ($\mathbf{B}_3$); and eventually, adding a differential entropy
term in meta loss to $\mathbf{B}_3$, resulting in our **Proposed** model.

We tested the five methods ($\mathbf{B}_0$–$\mathbf{B}_3$ and **Proposed**) on 10D and 20D Rastrigin functions with the same settings as in
Section 4.2. We compare the function minimum values returned by these methods in the table below (mean $\pm$ standard
deviation over 100 runs, each using 1000 function evaluations).

| Dimension | $\mathbf{B}_0$ | $\mathbf{B}_1$ | $\mathbf{B}_2$ | $\mathbf{B}_3$ | **Proposed** |
|---|---|---|---|---|---|
| 10 | 55.4±13.5 | 48.4±10.5 | 40.1±9.4 | 20.4±6.6 | 12.3±5.4 |
| 20 | 140.4±10.2 | 137.4±12.7 | 108.4±13.4 | 48.5±7.1 | 43.0 ±9.2 |

27
Our key observations are as follows. i) $\mathbf{B}_1$ v.s. $\mathbf{B}_0$: their performance gap is marginal. As suggested by R2, this proves
that our performance gain is not "just from having $k$ independent runs"; ii) $\mathbf{B}_2$ v.s. $\mathbf{B}_1$ and $\mathbf{B}_3$ v.s. $\mathbf{B}_2$: Whereas including
intra-particle attention in $\mathbf{B}_2$ already notably improves the performance compared to $\mathbf{B}_1$, including population-based
features and inter-particle attention in $\mathbf{B}_3$ presents the largest performance boost. This confirms that our method to
majorly "benefit from the attention mechanisms"; iii) **Proposed** v.s. $\mathbf{B}_3$: adding entropy from the posterior gains
further, thanks to its balance of exploration and exploitation. **We hope that the ablation study adds to a "thorough**
**experimental evaluation" and convinces R2 better.**

**Contribution to the ML field (R1).** We respectively disagree that our work was " a fairly straightforward combination
of optimizer learning and population-based optimization". Our work, for the first time, tackles a **novel and important**
**ML topic** (meta learning for population-based optimization) that leads to solving very rugged non-convex optimization
problems. Moreover, we believe **our methodology to be highly innovative and have broad implications** to other
topics in optimization and learning. First, an important complicacy in population-based optimization lies in the
collaboration among particles, which also presents a bottleneck when extending current point-based meta-optimizers.
We pioneered to address the bottleneck via the novel inter-particle attention mechanisms across LSTMs. Second, an
entropy term in meta loss, based on the posterior directly over the optimum, was designed to balance exploration and
exploitation, which is also " *a problem in other state-of-the-art (meta learning) approaches*" (Quote R2). Each of those
components contribute substantially, as shown in our ablation study above.

**We hope the above clarification has convinced R1 of our notable ML methodology innovations. In fact, we**
**notice the other two reviewers agreed on our work's significance and novelty**. Quoting R2: *"I consider that the*
*contributions of their work are novel, especially the proposed architecture" "the contributions are significant"*, and R3:
*"Their work opens the door to solving more sophisticated optimization using L2L".*

**Comparison to (Chen et al, 2017) and References (R3).** Although no official codes are available, we re-implemented
(Chen et al, 2017) and found its performance comparable to $\mathbf{B}_0$ (Andrychowicz et al., 2016), possibly due to their
similar model architectures. We will add references on attention mechanisms from the computer vision community.

[Meta-Review · NeurIPS 2019]

Good paper on on population-based meta optimizer in order to overcome the limitations of point-based and uncertainty-unaware current meta optimizers. The idea is novel and its application (on protein docking) makes additional solid contribution. Reviewers found that the paper was well written and easy to read. The reviewers appreciated the ablation study and clarifications in the rebuttal, so please revise the final draft accordingly